# Comparative Transcriptome Analysis Identifying the Different Molecular Genetic Markers Related to Production Performance and Meat Quality in *Longissimus Dorsi* Tissues of MG × STH and STH Sheep

**DOI:** 10.3390/genes11020183

**Published:** 2020-02-10

**Authors:** Shuru Cheng, Xueying Wang, Quanwei Zhang, Yuqin He, Xia Zhang, Lei Yang, Jinping Shi

**Affiliations:** 1College of Animal Science and Technology, Gansu Agricultural University, Lanzhou 730070, China; yanglei52hz@163.com (L.Y.); shijinpingyunfei@163.com (J.S.); 2College of Veterinary Medicine, Gansu Agricultural University, Lanzhou 730070, China; wangxydd2018@163.com; 3College of Life Science and Biotechnology, Gansu Agricultural University, Lanzhou 730070, China; heyq@gsau.edu.cn (Y.H.); zhangx@gsau.edu.cn (X.Z.)

**Keywords:** crossbred, genetics, meat quality, production performance, *Ovis aries*, transcriptome

## Abstract

Crossbred sheep have many prominent traits, such as excellent production performance and high-quality meat, when compared to local sheep breeds. However, the genetic molecular markers related to these characteristics remain unclear. The crossbred MG × STH (small-tailed Han sheep (STH) × Mongolian sheep (MG)) breed and the STH breed were selected to measure production performance and meat quality. We used 14 indexes of production performance and meat quality, which in the MG × STH population showed significant differences compared to the STH breed. Subsequently, the *longissimus dorsi* from the two sheep were subjected to comparative transcriptomic analyses to identify differentially expressed genes (DEGs) related to production performance and meat quality. A total of 874 DEGs were identified between the two sheep groups. A total of 110 unique DEGs related to sheep production performance and meat quality were selected as the candidate DEGs. We found 6 production-performance-related and 30 meat-quality-related DEGs through a correlation analysis, including *SPARC*, *ACVRL1*, *FNDC5*, and *FREM1*. The expression levels of 11 DEGs were validated by real-time PCR, and the results were in accordance with the results of the comparative transcriptomic and correlation analyses. These results will assist in understanding sheep heterosis and molecular marker-assisted selection.

## 1. Introduction

Mutton is an important protein source for humans and accounts for a large percentage of meat consumption in China [1]. With improvements in human living standards, the demand for high-quality meat and nutrition has also increased. In recent years, the production of mutton has steadily increased in China, but the increasing demand for high-quality meat mutton has not yet been satisfied [2]. Therefore, increasing the production performance and improving the meat quality of sheep breeds are pressing issues. Small-tailed Han sheep (STH) and Mongolian sheep (MG), two indigenous sheep breeds famous for their high-quality meat and strong adaptability, are farmed widely in northwestern China. However, there are significant differences between the STH and MG sheep breeds in terms of growth rate, production, and reproductive performance [2,3]. In general, the production performance and meat quality of STH sheep surpass those of MG sheep. Crossbreeding local sheep breeds with specialized breeds is the most common method used to rapidly improve the growth rate, meat quality, and production performance of a local sheep breed [4]. This crossbreeding method has been utilized to develop some excellent local breeds, such as Qianhua Mutton Merino (QHMM) sheep [5] and large-tailed Han (LTH) sheep [6]. Understanding the mechanisms of crossbreeding from a genetic perspective is important for optimizing sheep production performance and meat quality traits. The main purpose of this study was to identify the molecular markers related to production performance and meat quality for the further molecular breeding of sheep. However, the genes that regulate production performance and meat quality in crossbred sheep and parent breeds remain unclear.

Transcriptome analysis, a comprehensive sequencing technology that maps sequencing reads to the genome, can identify the genetic variants related to complex phenotypes [7]. Comparative transcriptome analyses of different pure sheep breeds such as STH sheep [8], Dorper sheep [9], Dorset sheep [10], and Mongolian sheep [11] have been performed and have detected some important molecular marker-assisted selection (MAS). For instance, calpastatin (*CAST*), myostatin (*MSTN*), and fatty-acid-binding protein 4 (*FABP4*) have been confirmed to be significantly associated with the production performance and meat quality of various sheep breeds [12]. However, comparative transcriptome analyses to identify the genetic differences between crossbred sheep and pure sheep, especially indigenous breeds such as QHMM sheep [3], LTH sheep [11], and Chinese Tan Sheep [13], have been reported only rarely. Comparative transcriptome profiling has identified *MRFs*, *GXP1*, and *STAC3* as likely being associated with muscle growth and development in QHMM sheep [3]. Comparative transcriptome analyses have also identified some important differentially expressed genes (DEGs), particularly metallothionein 3 (*MMP3*), related to a curly fleece in Chinese Tan sheep [13]. Overall, the genome-wide screening of crossbred and indigenous sheep breeds has identified some specific trait-associated genetic molecules, including *MYOD1*, *MYOG*, and *IGFBPs* [1,3,14]. However, understanding of the differences between the phenotypic and genetic aspects of different sheep breeds remains limited.

The association between phenotypes and genotypes in many livestock species is the result of precisely orchestrated molecular regulation networks [15], especially those markers associated with muscle growth, development, and muscle myofibers (myosin, troponin, myoglobin, and tropomyosin) [5,9],which are the dominant factors determining livestock production performance and meat quantity. In the present study, MG sheep were crossbred with STH sheep. Comparative transcriptome profiling in the *longissimus dorsi* tissues of the two sheep (the MG × STH and STH sheep) was conducted and combined with the production performance and meat quantity of these sheep to identify important phenotypic traits associated with DEGs. By understanding the genetic and phenotypic differences between MG × STH and STH sheep and by identifying precise MAS in MG × STH and STH sheep, the present study will extend the knowledge base of molecular breeding markers and provide novel candidate regulators for future genetic and molecular studies on optimizing sheep production and meat quality traits.

## 2. Materials and Methods

### 2.1. Sample Preparation, Collection, and Measures of Production Performance

Male MG sheep (*n* ≥ 50) mated naturally with indigenous female STH sheep (*n* ≥ 250) at a rate of 1:5 in Zhangye City (Gansu Province, China). Crossbred lambs (MG *×* STH) and STH sheep (females, *n* = 50, age, 45 day old) with similar birth ages (within 3 days) were selected for the present study. The lambs were complementarily fed with commercialized lamb milk replacer (ZL201210365927, Beijing Precision Animal Nutrition Center, China) and weaned at two months of age. All lambs were raised in a warm shed pen and in a lairage (space, 1 *×* 2.5 m; temperature, 10 to 17 °C) with regular feeding (dietary formula shown in Appendix A) three times per day (6:00 a.m.,12:00 a.m., and 18:00 p.m.) and free access to drinking water for six months, guaranteeing similar breeding and management conditions. The sheep fasted for 12 h before slaughter. The production performance and meat quality of these sheep were measured as described previously [3,16]. Briefly, live weight measurements were taken before slaughter. Electronic scales and tapelines were used to measure the slaughter performance metrics after 6 months of age, including body weight, body size, and carcass yield. Carcass percentage, dressing percentage, net meat percentage, and loin eye area were recorded in accordance with previously reported studies [17,18]. The meat quality was evaluated by shear force, pH, color chroma, and the muscle marbling grade of the longissimus dorsi tissue, in accordance with previously reported studies [2,19]. Samples from each group (females, *n* = 6; age, 6 months old) were randomly selected for transcriptome analysis. The samples were obtained immediately after slaughter and stored at −80 °C. All samples were collected in strict accordance with the ethical guidelines approved by the Animal Care Commission of the College of Animal Science and Technology, Gansu Agriculture University (GSAU-AEW-2017-0308).

### 2.2. RNA Isolation, Complementary DNA Library Construction, and Transcriptome Sequencing

Total RNA was extracted from the sheep longissimus dorsi tissues using Trizol reagent (TinaGen, Beijing, China). The RNA quality and quantity were assessed using a Nanodrop 2000 Spectrophotometer and an Agilent 2100 Bioanalyzer (Agilent Technologies, Sangta Clara, CA, USA). RNA samples of enough quality and quantity (RNA Integrity Number, RIN > 8.5) were enriched and fragmented using a Ribo-Zero™ Magnetic Kit (Epicentre, San Diego, CA, USA) according to the manufacturer’s protocol. The highest-quality RNA samples were selected for library construction, which were completed by Guangzhou Sagene Bioinformation Technology Co., Ltd. The cDNA libraries were constructed as described previously [20,21,22]. Briefly, cDNA and second-strand cDNA were synthesized and purified using a QiaQuick PCR kit (Qiagen, Redwood City, CA, USA) and ligated onto Illumina sequencing adapters. Then, 200–500-bp ligation products were selected using agarose gel electrophoresis, PCR-amplified, and sequenced using an Illumina HiSeq™2500 platform (Illumina, San Diego, CA, USA).

### 2.3. RNA-Sequencing Data Analysis

Raw reads were filtered to remove reads containing adapters, and the clean reads that passed quality control were mapped onto the sheep genome (*Oar_rambouillet_v1.0*) using Bowtie2 (v2.2.9) and TopHat2 (v2.1.1) [23,24]. The absolute value of the log2 (fold change), using fragments per kilobase million (FPKMs) [25] >1, was used as the threshold to determine the significantly differentially expressed genes (DEGs). The edgeR package [26] was utilized to identify the DEGs (criteria: fold change ≥ 1 and *p* < 0.05) with a false discovery rate (FDR) < 0.05 compared to the STH sheep group. The DEGs were used for gene ontology (GO) enrichment [27] and Kyoto Encyclopedia of Genes and Genomes (KEGG) pathway analysis [28]. The GO terms and pathways with *p* < 0.01 and *Q* < 0.01 were considered to be significantly enriched. We selected the DEGs from the GO terms and pathways that included the keywords “development”, “growth”, “muscle”, “fiber”, “myofibril”, “myoblast”, and “fat” (the potential targets involved in sheep production performance and meat quality, according to previous studies) [1,3,9,19]. A correlation analysis [29,30] between the significantly different indexes of the production performance and meat quality (*p* < 0.001) and the DEGs was conducted using the edgeR package with generalized linear mixed models (GLMMs) [31] to identify the multiple-trait- and trait-associated DEGs in the two sheep. The Pearson correlation [32] for candidate DEGs was calculated to construct a correlation coefficients matrix for further investigation and to determine the significance of co-expressed mRNA. The DEGs with Pearson correlation coefficient thresholds >0.75 for positive correlations or <−0.75 for negative correlations were considered to be targets for construction of the gene coexpression network using Cytoscape software [33]. A heatmap and a Venn diagram were drawn using edgeR to observe the differences in the DEGs.

### 2.4. Quantitative Real-Time PCR Assays for Target Genes

Total RNA was extracted using Trizol reagent (Invitrogen, Carlsbad, CA, USA) from the *longissimus dorsi* tissue samples of the MG × STH and STH sheep (*n* = 6). Complementary DNA was synthesized using a BioTeke Thermo RT Kit (Bioteke, Beijing, China) following the manufacturer’s instructions. The mRNA expression levels of 11 DEGs were validated for the RNA-seq and bioinformatics analysis data by quantitative real-time PCR (qPCR). The qPCR was performed using 2 μL of cDNA in a 25-μL reaction volume on an ABI7300 real-time system (ABI Systems, Foster City, CA, USA). SYBR Premix Ex Taq™ II and specific primers were used in each reaction. The primers were designed by Primer Premier 6.0 software [30] (Appendix A) and synthesized by Beijing Ubiolab Genetics Technology Co., Ltd. (Beijing, China). The expression of the housekeeping gene glyceraldehyde-3-phosphate dehydrogenase (*GAPDH*) was used as an intratissue control. The experimental process and the data analysis were carried out as described previously [34,35]. All RT-PCR reactions were performed in triplicate and included controls without a template.

### 2.5. Statistical Analysis of Production Performance in the MG × STH and STH Sheep

Statistical analyses of the production performance and meat quality in the MG × STH and STH sheep populations were performed using SPSS version 21.0 (SPSS Inc., Chicago, IL, USA). The data are expressed as the mean ± the SD, unless otherwise indicated. The production performance and meat quality, qPCR, and Western blot data were analyzed using *t*-test (between two groups) or one-way ANOVA analysis. The graphs were drawn using Prism version 5.0 (GraphPad Software Inc., La Jolla, CA, USA), and *p*-values less than 0.05 were considered statistically significant.

## 3. Results

### 3.1. Differences in Production Performance and Meat Quality between the MG × STH and STH Sheep

The production performance and meat quality of the MG × STH and STH sheep were evaluated using 19 indexes, and the results are shown in Appendix A. There were no significant differences between the two sheep among the six indexes related to production performance and meat quality, i.e., body height, body length, pH, water loss rate, lightness (L*), and redness (a*) (*p* ≥ 0.05). The remaining 13 indexes of the MG × STH sheep population had significant differences (*p* < 0.001 or *p* < 0.05) compared to the STH sheep breed (Figure 1). The chest circumference, live weight, carcass weight, and net meat mass of the MG × STH sheep were significantly higher than those of the STH sheep (Figure 1A–D). The marbling grade of the MG × STH sheep was higher than that of the STH sheep (*p* < 0.05), and the deposited fat (observed visually) of the MG × STH sheep was greater than that of the STH sheep (Figure 1E). The meat quality, cooking loss, and yellowness (b*) of the STH sheep were higher than those of the MG × STH crossbred sheep (Figure 1F,G), while the shear force (kgf) of the MG × STH sheep population was greater than that of the STH sheep (Figure 1H). These results suggest that the MG sheep crossbred with the STH sheep had improved production performance, but the meat quality was inferior to that of STH sheep.

### 3.2. Transcriptome Analysis of Longissimus Dorsi Tissues from the MG × STH and STH Sheep

A comparative transcriptome analysis of longissimus dorsi tissue was performed in order to investigate the differences, from a genetic perspective, between the MG *×* STH and STH sheep in terms of production performance and meat quality. An average of 9,847,408,899 and 10,278,304,671 clean reads were obtained from the MG *×* STH and STH sheep, respectively. After filtering of the redundant sequences, a total of 64,492,290 reads from the MG *×* STH sheep and 67,220,735 reads from the STH sheep were mapped onto the sheep reference genome for transcriptomic analysis using Cufflinks (Appendix A). The mapping ratio of each sample was greater than 73%. The reassembled transcript fragments were used to identify and statistically analyze the gene expression levels (Figure 2). A total of 22,556 genes, including 21,108 known and 1448 novel genes, were obtained from the MG *×* STH sheep and a total of 20,884 genes, with 19,740 known and 1141 novel genes, were annotated for the STH breed (Figure 2A). A total of 810 genes, including 541 up-regulated and 269 down-regulated genes with FDR < 0.05 and log_2_FC > 1, were identified as DEGs compared to the STH group (Figure 2B and Appendix A). The up-regulated DEGs were approximately twice as common as the down-regulated DEGs, and the majority of the up-regulated DEGs displayed fold changes > 2 (Figure 2C). Subsequently, a GO and pathway enrichment analysis was performed with these DEGs. A total of 173 GO terms, including 130 about biological processes, 23 about molecular functions, and 20 about cellular components (Figure 2D) were significantly enriched (*p* < 0.05). The pathway analysis identified 38 significantly different pathways (*p* < 0.05). Figure 2E shows the top 10 most enriched pathways, where most of the DEGs (*p* < 0.01), such as the PI3K-Akt signaling pathway and the regulation of actin cytoskeleton pathway, play crucial roles in development and the growth of muscle (Figure 2E).

### 3.3. Identification of the Candidate DEGs Related to the Production Performance and Meat Quality in MG × STH and STH Sheep

We focused on DEGs that affected sheep production performance and meat quality, as previously described (Figure 3). A total of 145 DEGs, including 25, 99, and 21 related to growth, development, and meat quality, respectively, were selected as the candidate DEGs (Figure 3A). A Venn diagram was constructed to filter the repeated DEGs (Figure 3B). Finally, we found a total of 110 unique DEGs related to sheep production performance and meat quality (Appendix A). Specifically, seven DEGs (*ADIPOQ*, *IFRD1, PLCB1, ITGB3*, *GPX1, HEG1*, and *HEYL*) were commonly expressed in the three biological processes, and these genes were also down-regulated with a -log2 (fold change) from 1 to 2.5 (compared to the STH sheep breed) (Figure 3C). A heatmap was constructed using these 110 DEGs (Figure 2D). Most of the candidate DEGs showed a significant difference between the MG × STH and the STH sheep. However, some DEGs did not show significant differences when visualized via the heatmap.

### 3.4. Correlation Analysis of Multiple Traits and Trait-Associated DEGs in MG × STH and STH Sheep

In order to better identify and confirm the candidate DEGs related to production performance and meat quality traits (Figure 4), we calculated the correlation between the DEGs and eight significantly different traits (*p* < 0.001). A gene–trait relationship heatmap was established according to Pearson correlations and the 110 unique DEGs (Figure 4A). The results showed that the DEGs with Pearson correlation coefficient thresholds >0.75 or <−0.75 were related to chest circumference (6 DEGs), carcass weight (1 DEG), net meat mass (1 DEG), yellowness (b*) (13 DEGs), cooking loss (12 DEGs), and shear force (6 DEGs) (Appendix A). Subsequently, a Sankey diagram was created to reflect the relationship between the DEGs and the significantly different traits (Figure 4B). There were some DEGs related to one or more production performance and meat quality traits. For example, *SPARC*, *FNDC5*, and *ACVRL1* were simultaneously associated with two or three traits, whereas *ADIPOQ* and *NMRK2* were only associated with one trait. We also compared the expression of these genes between the two sheep based on their Log (FPKM) values, and the results showed that these genes were expressed differentially between the two sheep (Figure 4C,D).

### 3.5. Gene Coexpression Analyses of Targeted DEGs Related to Production Performance and Meat Quality in MG × STH and STH Sheep

In order to obtain highly dependable candidate DEGs related to production performance and meat quality in the MG × STH and STH sheep, the DEGs with Pearson correlation coefficient thresholds >0.75 or <−0.75 were considered to be targets for further analysis (Figure 5). We found 6 and 30 candidate DEGs related to production performance and meat quality, respectively (Figure 5A). Notably, four DEGs (*SPARC*, *ACVRL1*, *FNDC5*, and *FREM1*) were related both to production performance and to meat quality. These genes showed a significant difference between all samples (Figure 5B). In addition, in the co-expression network (Figure 5C), 27 of the 30 DEGs were discovered and mutually regulated, particularly *SPARC*, *ACVRL1*, *NMRK2*, and *COL3A1*.

### 3.6. Validation of the DEGs Identified from Transcriptome and Associated Trait Correlation Analyses

The mRNA expression levels of six DEGs (*IFRD1*, *PLCB1*, *ITGB3*, *GPX1*, *HEG1*, and *HEYL*) expressed commonly in the two sheep that were related to growth, development, and meat quality were selected for validation of the transcriptomic data. Another five DEGs (*ADIPOQ*, *SPARC*, *FNDC5*, *ACVRL1*, and *NMRK2*) were selected for validation of the correlation analysis data using qPCR (Figure 6). Compared to the STH sheep, the six DEGs identified from the transcriptome analysis were significantly down-regulated in the MG × STH sheep (Figure 6A). The expression levels of these genes, determined by qRT-PCR, were consistent with the transcriptome data. The additional five DEGs selected from the correlation analysis also were significantly down-regulated in the MG × STH crossbred sheep (Figure 6B). *NNMRK2* was the sole exception, as it was up-regulated in the MG × STH sheep. The significant expression levels of *SPARC* and *NNMRK2* not only proved the accuracy of the correlation analysis data, but also demonstrated the veracity of the gene co-expression network. The verified results from the qPCR had trends similar to the transcriptomic results, suggesting that these DEGs could play important roles in the regulation of production performance and meat quality in sheep.

## 4. Discussion

Genotypes are the main dictators of lamb growth potential, growth rate, and development, and they also determine muscle mass and fat deposition [36]. Understanding the differences in genotypes is important for optimizing production performance and meat quality in different sheep breeds. The phenotype differences between MG × STH and STH sheep were evaluated, and the results suggest that MG × STH sheep are beneficial for enhancing sheep production performance but not improving meat quality. Compared to previous data on production performance and meat quality regarding other sheep breeds, such as QHMM and STH sheep [3,37], these data, for MG × STH and STH sheep, were significantly different. This is because genotype can be influenced by environmental and nutritional factors [36], which is why the mutton of northwestern China is superior to that of other regions.

To identify the DEGs related to differences between MG × STH and STH sheep in terms of production performance and meat quality, a comparative transcriptome analysis of the *longissimus dorsi* muscles was performed. Compared to the STH group, a total of 810 DEGs were identified in the STH × MG sheep. This number is slightly less than that reported for some crossbred sheep but greater than that reported for commercial and indigenous sheep [3,6,10]. This is the advantage of crossbreeding different sheep, which can increase economic performance and genetic resources. On the other hand, identifying the DEGs in different crossbred sheep is more valuable in terms of MAS. Subsequently, the results of the GO and pathway enrichment analysis showed some important DEGs related to the reproduction of sheep, which was in accordance with other observations that the reproduction performance of STH × MG sheep is superior to that of MG sheep [2,11]. Moreover, some DEGs are related to the immune system and are a response to environmental adaptation, which are some of the advantages of crossbred sheep [38]. However, we focused on the DEGs related to production performance and meat quality in STH × MG and STH sheep. A total of 110 unique DEGs related to growth, development, and meat quality were identified (in accordance with selection principles in previous studies) [3,9,39]. Specifically, the seven most commonly expressed DEGs (*ADIPOQ*, *IFRD1, PLCB1, ITGB3*, *GPX1, HEG1*, and *HEYL*) showed significant differences between the two sheep. Previous studies have also indicated that *GXP1* [3], *ADIPOQ* [40], and *IFRD1* [41] are essential for muscle growth and development, which was reliably confirmed in our results. In order to identify the DEGs precisely related to the phenotype differences between STH × MG and STH sheep, the correlation was analyzed, and the results showed that 6 DEGs were related to production performance and 26 DEGs were related to meat quality, particularly *SPARC*, *ACVRL1*, and *ADIPOQ* (*R* > 0.80). One previous study has reported that *SPARC* and *ACVRL1* play important roles in the proliferation and development of endothelial cells [42], suggesting that these DEGs could be targets affecting sheep production performance and meat quality. It has been confirmed that the GLMM approach can be applied as a convenient means of testing the correlations between multiple traits and DEGs using a single model relating gene expression to caste phenotype and species traits [31]. In addition, a Sankey diagram was used to reflect the correlation between DEGs and phenotypes, and the results suggested some important DEGs, such as *SPARC*, *ACVRL1*, *NMRK2*, and *COL3A1*. These results were also verified by the co-expression network analysis, which showed that 27 of the 30 DEGs were discovered and mutually regulated. However, three novel DEGs (*LOC101122119*, *TCONS_00068308*, and *LOC101112606*) were not present in the network, which means that these genes are not important for sheep production performance or meat quality. On the contrary, these DEGs showed a high correlation with cooking loss, such as *LOC101112606* (0.92) and *LOC101122119* (0.76). These results suggest that the function of these genes still needs to be further clarified. Finally, we chose some DEGs to validate the transcriptome and correlation analysis data using qPCR, which showed the reliability of our data. It should be noted that we found *NMRK2* to be the most significantly DEG. This was supported by the results of the coexpression network, which showed that *NMRK2* was the gene with the most regulatory nodes. To summarize, our findings confirm that 32 DEGs affect 8 significantly different traits in MG × STH and STH sheep. However, further studies are required to investigate the roles of these candidate genes in genetic breeding and in improving the production performance of sheep and other livestock.

## 5. Conclusions

In the present study, we studied the phenotype differences and comparative transcriptome in production performance and meat quality between MG × STH and STH sheep. The results demonstrate that production performance in MG × STH sheep is significantly superior to that of STH sheep, while the meat quality of MG × STH sheep is inferior to that of STH sheep. Finally, we found 6 production-performance-related and 30 meat-quality-related DEGs. These results will assist in understanding sheep heterosis and molecular marker-assisted selection.

## Figures and Tables

**Figure 1 genes-11-00183-f001:**
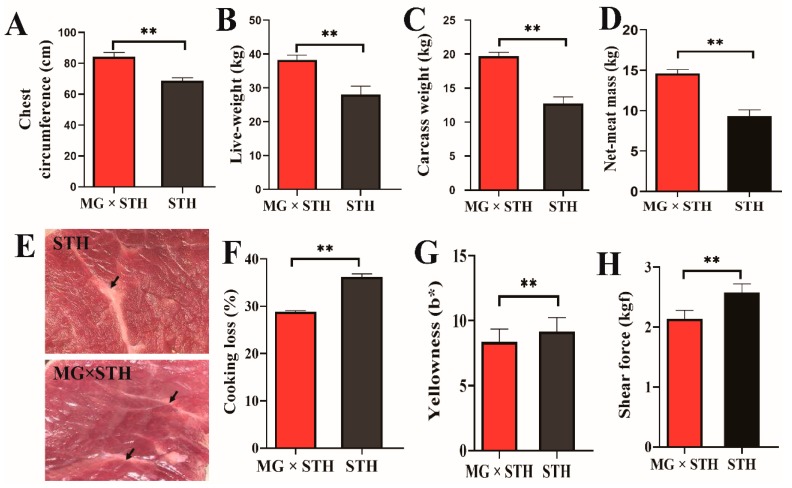
Significantly different indexes related to production performance and meat quality between the Mongolian (MG) × small-tailed Han (STH) sheep and the STH sheep. (**A****–D**) The indexes of chest circumference, live weight, carcass weight, and net meat mass used for evaluating the production performance of MG × STH and STH sheep. (**E**) The marbling of MG × STH and STH sheep. The black arrow shows the deposited fat in the marbling. (**F–H**) The indexes of cooking loss, yellowness (b*), and shear force used for assessing the meat quality of the MG × STH and STH sheep. ***p* < 0.001.

**Figure 2 genes-11-00183-f002:**
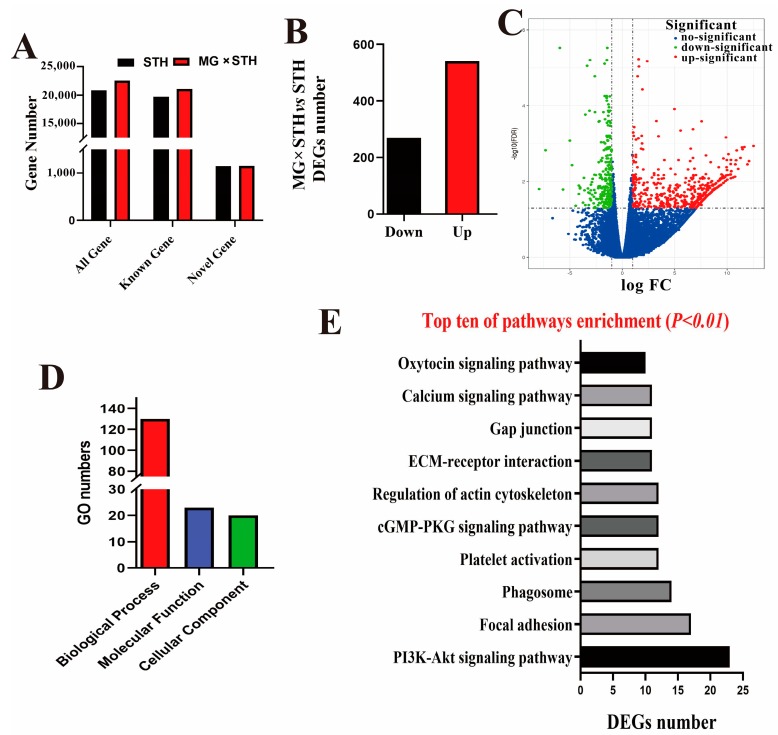
Identity and functional analysis of the differentially expressed genes (DEGs) from *longissimus dorsi* tissues in the MG × STH and STH sheep. (**A**) Total number of genes identified by transcriptome sequencing in the two sheep groups. (**B**) The number of DEGs compared to the STH sheep. (**C**) Volcano plot of the DEGs. (**D**) The significant gene ontology (GO) numbers of cellular components, molecular functions, and biological processes (*p* < 0.05). (**E**) Top 10 most significant pathways according to the DEGs in the two sheep groups (*p* < 0.05), where the *x* axis represents the number of DEGs.

**Figure 3 genes-11-00183-f003:**
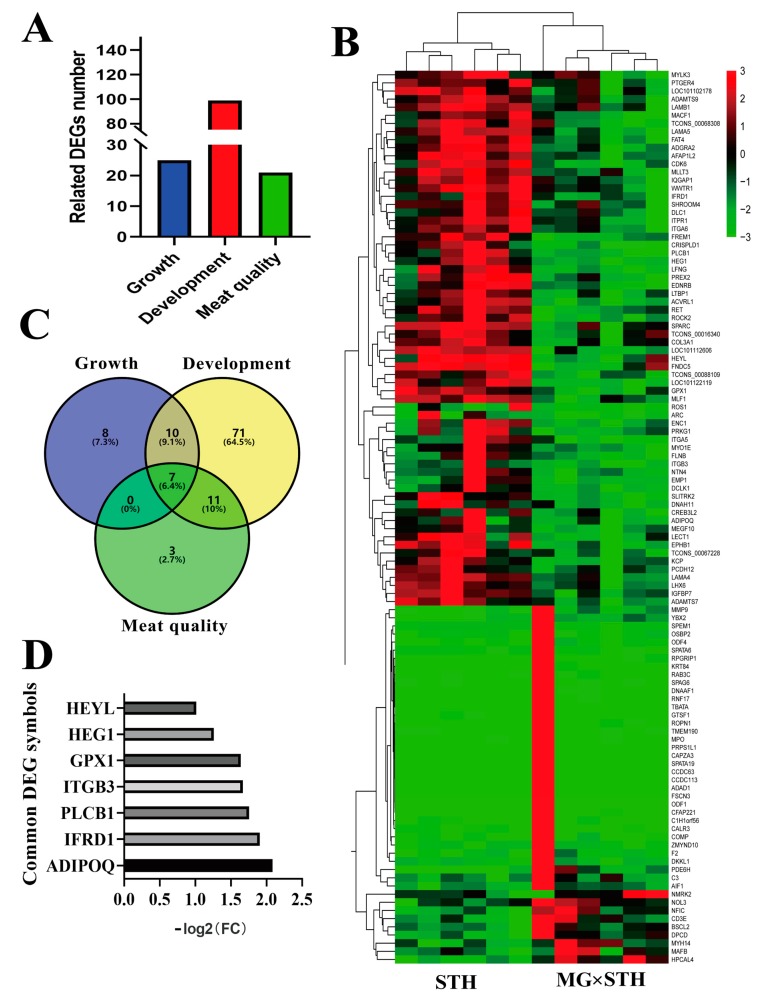
Identification of candidate DEGs related to production performance and meat quality in the MG × STH and STH sheep. (**A**) Identification of the candidate DEGs related to growth and development in the two sheep groups. (**B**) A heatmap of the DEGs related to growth, development, and meat quality in each group. (**C**) A Venn diagram of DEGs related to growth, development, and meat quality. (**D**) The common DEGs involved in growth, development, and meat quality and their relative expression using a transcriptomic analysis. The *x* axis represents the relative expression by −log_2_(Fold Change).

**Figure 4 genes-11-00183-f004:**
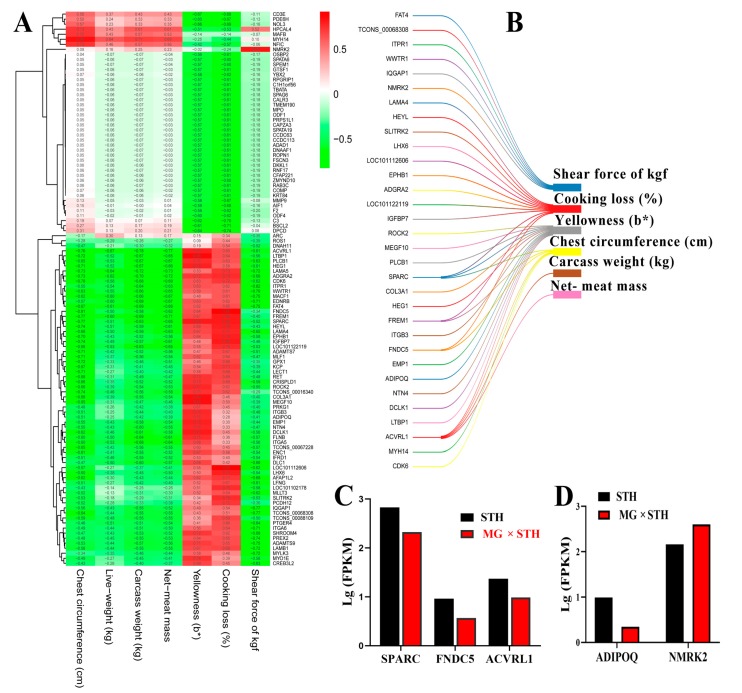
A correlation analysis between DEGs and production performance and meat quality traits. (**A**) A heatmap of the gene–trait relationship established according to the Pearson correlation of DEGs and significantly different production performance and meat quality traits. (**B**) A Sankey diagram created with Pearson correlation coefficient thresholds >0.75 or <−0.75. (**C**,**D**) The relative expression of five candidate DEGs from RNA-seq data, where the expression value was calculated according to the log(FPKMs). FPKMs: Fragments per kilobase million.

**Figure 5 genes-11-00183-f005:**
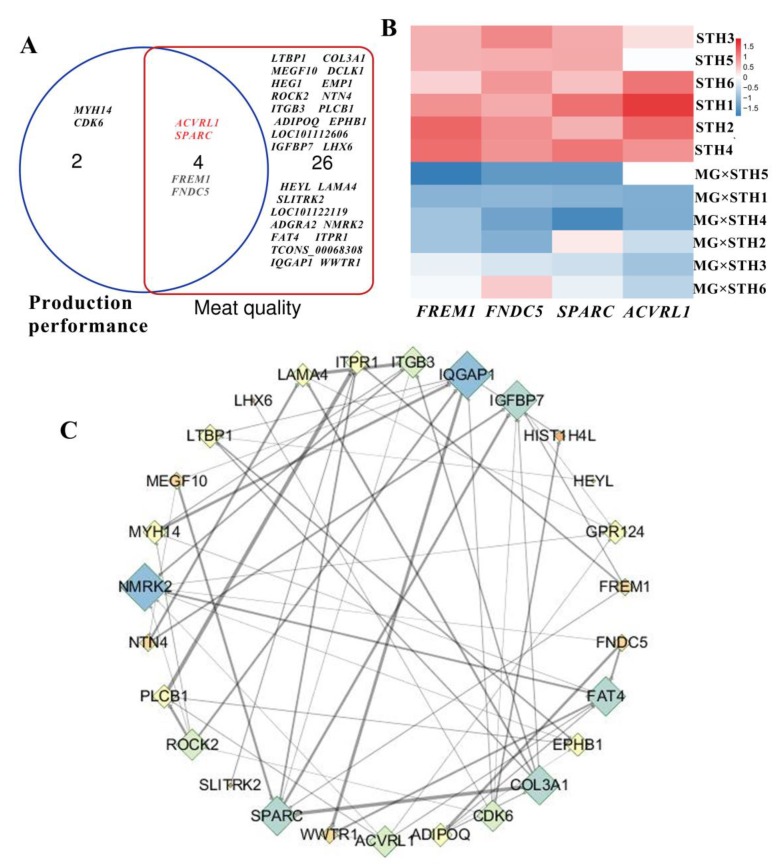
Gene co-expression analyses of targeted DEGs related to sheep production performance and meat quality. (**A**) A Venn diagram of DEGs with Pearson correlation coefficient thresholds >0.75 or <−0.75 related to production performance and meat quality. (**B**) A heatmap of four DEGs related both to production performance and to meat quality. (**C**) The gene co-expression network of targeted DEGs (>0.75 or <−0.75) related to sheep production performance and meat quality.

**Figure 6 genes-11-00183-f006:**
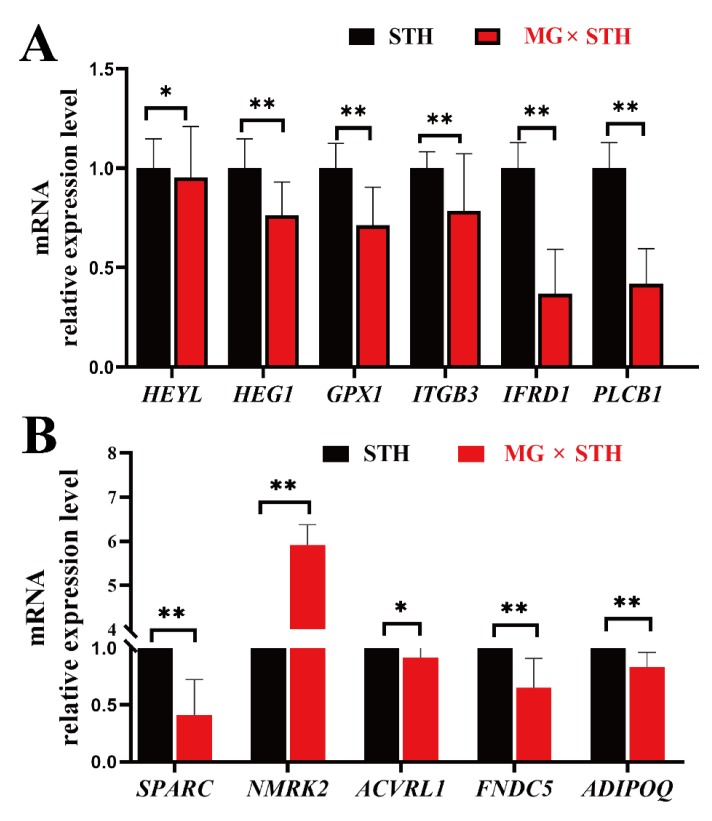
Validation of 11 target DEGs in terms of transcriptome and correlation analysis data. (**A**) qRT-PCR analysis of six DEGs validated for transcriptome data. (**B**) qRT-PCR analysis of five DEGs validated for trait-associated correlation analysis data. The values were calculated using the 2^−ΔΔCt^ method: * *p* < 0.05, ** *p* < 0.01.

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
