# Peer review of "Comparative Transcriptome Analysis Identifying the Different Molecular Genetic Markers Related to Production Performance and Meat Quality in Longissimus Dorsi Tissues of MG × STH and STH Sheep"

_genes, 2020, doi:10.3390/genes11020183_

Round 1
Reviewer 1 Report
Dear authors
This is an interesting paper and will add value to the field of sheep genomics. I have a few comments in terms of the materials and methods section on the sheep husbandry.
Line 87 - This is confusing, what was the sample number of lambs used for phenotypic recording, 30 or 50 female lambs? And why were you using females and not males?
Line 88 states that the lambs were fed calf starter feed. It does not make sense to feed lambs calf starter unless if the authors are referring to creep feed. Please indicate the composition of the creep feed. Secondly, the authors use the word ' quantitative feed ' which is technically not the correct terminology - can this be corrected? Did the lambs receive an adlib formulated diet, how was the quantity of the diets given determined and for how long were the lambs fed before slaughter?
It would also be of benefit to include the environmental conditions in this region of China. Also, the housing conditions of the lambs are unclear. Were they kept individually in pens or in groups? How much space did they have?
Lines 143, 489, 501-503 - font size is different than the rest of the text, please keep this consistent.
Line 160 - should rather be results and discussion
Line 165 - spelling of remaining
line 166 - If P<0.05 then it is significantly different
line 477 - Section 4 should be the conclusion of the article
line 479 - spelling of differences
line 515 - spelling of DEG
line 516 - write out GLMM in full
line 525-526 - Sentence should be rephrased to read better
The font for the references is not according to the template, the authors should check the references as there are journal names that are missing.
Author Response
Reviewer 1:
Reply: Thanks for your review. Here is the point to point response to the reviewer’ comments.
Line 87 - This is confusing, what was the sample number of lambs used for phenotypic recording, 30 or 50 female lambs? And why were you using females and not males?
Reply:The phenotypes of female MG and STH including the body weight, body length and body height are more closed to each other. For example, the body weight of female MG and STH is 54.20 kg and 57.30 kg, respectively. However, the body weight of male MG (69.70 kg) and STH (80.50 kg) are significantly different. In addition, compared to MG and STH sheep breed, the crossbred of male MG×STH also showed some significantly difference. In order to identify the molecular markers related to production performance and meat quality for further MAS of sheep, we need selected the individuals that have approximately phenotypes. So, in present study, the female sheep was selected for production performance and meat quality analysis and the comparative transcriptome analysis.
Line 88 states that the lambs were fed calf starter feed. It does not make sense to feed lambs calf starter unless if the authors are referring to creep feed. Please indicate the composition of the creep feed. Secondly, the authors use the word ' quantitative feed ' which is technically not the correct terminology - can this be corrected? Did the lambs receive an adlib formulated diet, how was the quantity of the diets given determined and for how long were the lambs fed before slaughter? It would also be of benefit to include the environmental conditions in this region of China. Also, the housing conditions of the lambs are unclear. Were they kept individually in pens or in groups? How much space did they have?
Reply:Thank you for give us some important suggestions. We are sorry about this. Here, we used a commercialized the lamb milk replacer (ZL201210365927, Beijing. China) before weaned at 2 months of age. All lambs were raised in a warm shed pen and in a lairage (space, 1 m × 2.5 m, temperature, 10 to 17℃) with regular feeding (dietary formula show in Supplementary Table S1) with three times per day(6: 00 am,12:00 am and 18:00 pm) and free access to drinking water for 6 months, guaranteeing similar breeding and management conditions. Feed was fast for 12 h before slaughter.
Lines 143, 489, 501-503 - font size is different than the rest of the text, please keep this consistent.
Reply: We revised all these mistakes throughout the manuscript.
Line 160 - should rather be results and discussion
Reply: Section 3 is the results. We revised some parts based on your comments.
Line 165 - spelling of remaining
Reply: We revised all these mistakes throughout the manuscript.
line 166 - If P<0.05 then it is significantly different
Reply: Here, because the P>0.05, so these indexes are no significant difference.
line 477 - Section 4 should be the conclusion of the article
Reply: We have revised this section and added the conclusion section in our manuscript.
line 479 - spelling of differences
Reply: We revised all these mistakes throughout the manuscript.
line 515 - spelling of DEG
Reply: We revised all these mistakes throughout the manuscript.
line 516 - write out GLMM in full
Reply: We have added the full style of the GLMM in Line 130 that first used.
line 525-526 - Sentence should be rephrased to read better
Reply: We revised it based on your comments and the content of this sentence.
The font for the references is not according to the template, the authors should check the references as there are journal names that are missing.
Reply: We have carefully checked and revised it.

Reviewer 2 Report
The paper reports the transcriptome analysis in sheep to look for relationships with meat production and quality. The investigation has been well planned and carried out and the paper is well organized and clearly presented. Therefore I only have minor comments.
Comments:
L 5: it would be better to avoid (in the whole text) the definition ‘sheep breed’ when it refers also to MGxSTH, because this is not a breed
L 16: ‘hybrid’ is uncorrect because the animals derived from a cross between two sheep breeds à replace with ‘crossbred’, throughout the entire paper
L 46: in the most -> is the most
L 47: growing speed -> growth rate
L 52: delete ‘vital’. It is unclear and unnecessary
L 54:crossbreeding -> crossbred
L 56: complicated -> complex
L 59: obtained -> detected
L 81: what a genetic ‘breeding’ molecular marker is?
L 87: usually the sire breed is indicated first -> MG x STH
L90: ‘consistent’ breeding conditions à same/similar ???
L91: Here the age at slaughter should be indicated
L 105: accessed -> assessed
L 109: the higher -> the highest
L 139: total RNA were -> total RNA was
L 150: All PCR -> all RT-PCR
L 164: PH -> pH
L 165: brightness value -> lightness (L*); red value -> redness (a*). The * is necessary if the CieLab system was used
L 166: the differences (P<0.05) are also significant, as stated in L 158-159. The same in table S2.
L 168: what ‘net-meat weight’ means? Is the carcass weight in fig. 1C?
L 171: moisture cooking loss -> the correct term is simply ‘cooking loss’. Throughout the text
L 172: there is no reason for reporting the shear force in two unit of measure
L 174: delete ‘obviously’
L 175: actually the cooking loss in MGxSTH was lower, which is positive
L 240: A totaly -> a total
L 242: top ten -> in fig 2E nine pathways are shown
L337: there DEGs -> the DEGs
L 339: yellowness value (b) -> yellowness (b*)
L342: delete ‘some’
L 344: investigated -> compared
L 479: why maximum? The genotype could determine also the minimum
L 484: tenderness was lower in MGxSTH, but the other examined quality traits were similar, or even better (e.g. cooking loss). It is true that tenderness is one of the most important traits, but to say that the meat quality (which includes all the traits) is lower seems to me too much
L 484-486: it is not clear which kind of comparison was made with other breeds
L 489: the genotype was not investigated
L 495: It was also demonstrated…: the sentence doesn’t make sense in the context
L 498-500: it can be true, but the reproduction performance was not considered in the work. A reference was missed?
In general, the Discussion section is more a repetition of the results than a real discussion
References:
In most of the references the name of the journal is missing (and in one case also the name of the authors). This made it impossible to check the appropriateness of certain comments in the Results and especially in the Discussion section

Author Response
Reviewer 2:
The paper reports the transcriptome analysis in sheep to look for relationships with meat production and quality. The investigation has been well planned and carried out and the paper is well organized and clearly presented. Therefore I only have minor comments.
Reply: Thanks for your review. Here is the point to point response to the reviewer’ comments.
Comments:
L 5: it would be better to avoid (in the whole text) the definition ‘sheep breed’ when it refers also to MGxSTH, because this is not a breed
Reply: Thanks for your review. We have revised it in throughout the manuscript.
L 16: ‘hybrid’ is uncorrect because the animals derived from a cross between two sheep breeds à replace with ‘crossbred’, throughout the entire paper
Reply: Thank you for bringing up this point. We have revised it throughout the manuscript.
L 46: in the most -> is the most
Reply: We have revised it with red color based on your comments.
L 47: growing speed -> growth rate
Reply: We have revised it with red color based on your comments.
L 52: delete ‘vital’. It is unclear and unnecessary
Reply: We have revised it with red color based on your comments.
L 54:crossbreeding -> crossbred
Reply: We have revised it with red color based on your comments.
L 56: complicated -> complex
Reply: We have revised it with red color based on your comments.
L 59: obtained -> detected
Reply: We have revised it with red color based on your comments.
L 81: what a genetic ‘breeding’ molecular marker is?
Reply: Here we means Molecular Marker-assisted Selection (MAS). We have revised in our manuscript.
L 87: usually the sire breed is indicated first -> MG x STH
Reply: Thank you for bringing up this point. We have revised it throughout the manuscript.
L90: ‘consistent’ breeding conditions à same/similar ???
Reply: Here, we means the same or similar condition. We have changed the word and instead of it.
L91: Here the age at slaughter should be indicated
Reply: We have added the age at slaughter in manuscript.
L 105: accessed -> assessed
L 109: the higher -> the highest
L 139: total RNA were -> total RNA was
L 150: All PCR -> all RT-PCR
L 164: PH -> Ph
Reply L 105-L 164: Thank you for review so carefully. We have revised all these mistakes like these in our manuscript.
L 165: brightness value -> lightness (L*); red value -> redness (a*). The * is necessary if the CieLab system was used
Reply: Thank you for bringing up this point. We have revised it.
L 166: the differences (P<0.05) are also significant, as stated in L 158-159. The same in table S2.
Reply: Yes, the P<0.05 means significant differences. We have revised it based on your cmments.
L 168: what ‘net-meat weight’ means? Is the carcass weight in fig. 1C?
Reply: Here the net-meat weight should change as net-meat mass that has used in some studies. The net-meat weight (mass) is not the carcass weight. The carcass weight is including the bone weight however the net-meat weight (mass) is not including the bone weight. We have revised it in this part.
L 171: moisture cooking loss -> the correct term is simply ‘cooking loss’. Throughout the text
Reply: Thank you for bringing up this point. We have revised it throughout the manuscript.
L 172: there is no reason for reporting the shear force in two unit of measure
Reply: Thank you for bringing up this point. We reviewed some references that selected the Shear force (kgf) as an important index for evaluating the meat quality. Finally, we selected this index in our manuscript and revised it. Thank you very much for your meaningful suggestions.
L 174: delete ‘obviously’
Reply: We have revised it based on your comments.
L 175: actually the cooking loss in MGxSTH was lower, which is positive
Reply: We are sorry about this. This is not the cooking loss. We have carefully checked all indexes and revised these.
L 240: A totaly -> a total
Reply: We have revised it with red color based on your comments.
L 242: top ten -> in fig 2E nine pathways are shown
Reply: We are sorry about this. We have carefully checked and revised it.
L337: there DEGs -> the DEGs
Reply: We have revised it with red color based on your comments.
L 339: yellowness value (b) -> yellowness (b*)
Reply: We have revised it based on your comments.
L342: delete ‘some’
Reply: We have deleted it based on your comments.
L 344: investigated -> compared
Reply: We have revised it with red color based on your comments.
L 479: why maximum? The genotype could determine also the minimum
Reply: Previous studies used this word as follow. Animal growth is defined as an increase in bodyweight that is achieved by hypertrophy and hyperplasia until a mature size is reached, accompanied by changes in body conformation (Lawrie, 1998). Mature size is generally regarded as the point of maximum muscle mass at which the level of fat deposition increases (Owens et al., 1993). Here, we cited this reference that show the important of the genotype affecting the growth rate of lamb. We have revised it with red color based on your comments.
L 484: tenderness was lower in MGxSTH, but the other examined quality traits were similar, or even better (e.g. cooking loss). It is true that tenderness is one of the most important traits, but to say that the meat quality (which includes all the traits) is lower seems to me too much
Reply: have revised it based on your comments.
L 484-486: it is not clear which kind of comparison was made with other breeds
Reply: Thank you for bringing up this point. Here, we compared the data of production performance and meat quality in our study and those data that previous reported in other sheep such as QHMM and STH. We have revised it with red color based on your comments.
L 489: the genotype was not investigated
Reply: We have deleted the word (genotype) and revised this sentence with red color based on your comments.
L 495: It was also demonstrated…: the sentence doesn’t make sense in the context
Reply: We have deleted it based on your comments and carefully revised some redundant sentences.
L 498-500: it can be true, but the reproduction performance was not considered in the work. A reference was missed?
Reply: We revised it and added the reference based on your comments.
In general, the Discussion section is more a repetition of the results than a real discussion
Reply: We have revised it and rephrased this section.
References:
In most of the references the name of the journal is missing (and in one case also the name of the authors). This made it impossible to check the appropriateness of certain comments in the Results and especially in the Discussion section
Reply: we are sorry about this. In general, we used the endnote software and the template to conduct it, but it made some mistakes. We have checked it and revised all references. Thank you very much.

Reviewer 3 Report
The article: “Comparative transcriptome analysis identifies the 2 different molecular genetic markers related to 3 production performance and meat quality in longissimus dorsi tissues of two sheep breeds".
The aim of the presented research was to improve production characteristics and quality of mutton by crossbreeding Small Tailed Han x Mongolian sheep. The research consisted in comparing the transcriptome in the longest muscle in both breeds with respect to production characteristics and meat quality.
Notes to the manuscript:
Growth studies were carried out on 50 individuals of each breed, no indication of the sex was given. Likewise, no information was given on which sex was the transcript of the study, only that 6 individuals from each group were given. The description of mothers'; and lambs'; feeding is not precise and needs to be supplemented. The composition of the dose and its nutritional value was not given. Results
It is not possible to open additional files with tables with data on growth parameters and meat quality, so it is impossible to verify the results in the text. The Discussion chapter is written correctly, there is no comparison behind the research of other authors only references are to their previous research
References
The list of references contains a lot of errors and omissions and needs to be thoroughly checked and supplemented. The following items of literature do not contain the name of the magazine, among others in position: 2,3,4,5,6,8,9,10,11,12,13, 16,22,31,35, 41,45,46,47.
The work needs to be improved, in this form it is not suitable for printing in the “Genes".
Author Response
Revierwer 3:
Comments and Suggestions for Authors
The article: “Comparative transcriptome analysis identifies the 2 different molecular genetic markers related to 3 production performance and meat quality in longissimus dorsi tissues of two sheep breeds". The aim of the presented research was to improve production characteristics and quality of mutton by crossbreeding Small Tailed Han x Mongolian sheep. The research consisted in comparing the transcriptome in the longest muscle in both breeds with respect to production characteristics and meat quality.
Reply: Thanks for your review. Here is the point to point response to the reviewer’ comments.
Notes to the manuscript:
Growth studies were carried out on 50 individuals of each breed, no indication of the sex was given. Likewise, no information was given on which sex was the transcript of the study, only that 6 individuals from each group were given.
Reply: Indeed, we described it in the section 2.1. All of the 50 individuals are female and including the samples used for transcriptome analysis. In order to clarify the information of the samples, we revised this paragraph based on your comments.
The description of mothers'; and lambs'; feeding is not precise and needs to be supplemented. The composition of the dose and its nutritional value was not given.
Reply: Thank you for give us some important suggestions. We have revised this section based on your comments. The detail showed in section 2.1. Sample preparation, collection and measure of production performance.
Results
It is not possible to open additional files with tables with data on growth parameters and meat quality, so it is impossible to verify the results in the text.
Reply: We have revised all section based on all reviewers’ comments and uploaded it and the files again.
The Discussion chapter is written correctly, there is no comparison behind the research of other authors only references are to their previous research
Reply: We have rephrased this section. The details showed in manuscript with red color.
References
The list of references contains a lot of errors and omissions and needs to be thoroughly checked and supplemented. The following items of literature do not contain the name of the magazine, among others in position: 2,3,4,5,6,8,9,10,11,12,13, 16,22,31,35, 41,45,46,47.
The work needs to be improved, in this form it is not suitable for printing in the “Genes".
Reply: we are sorry about this. In general, we used the endnote software and the template to conduct it, but it made some mistakes. We have checked it and revised all references. Thank you very much.
Round 2
Reviewer 3 Report
The second version of the manuscript submitted for correction has been corrected according to the notes given to the first version of the article. Most of the comments have been taken into account.
If the lead Editor accepts this version, I'm inclined to follow his decision.
Author Response
Reply: Thanks for your review. We have carefully revised some mini-errors.
Academic Editor Comments
Notes for Authors: While authors have analysed 50 MG x STH cross-bred lambs and compared them to 50 STH lambs, they make no mention of how many sires were used to generate those lambs. This is important as the ram(s) used may well bias the phenotypic comparisons (Table S3 and manuscript), especially if the lambs studied are all closely related (i.e. progeny of a small number of sires). At the very least a sire correction should be used in the comparisons, especially as the authors do suggest that there is quite considerable sexual dimorphism in one of the breeds.
Author's Reply: Reply: Indeed, we have considered these situations. Male MG (n≥50) sheep were mating naturally with indigenous female STH sheep (n≥250) at the rate of 1:5. As the reviewer 1 mentioned in his/her reviews. Because the phenotypes of the female MG, STH and crossbred MG×STH including the body weight, body length and body height are more closed to each other. So, we used the female as our experiment object. In addition, the female sheep can give birth two or three lambs. In order to avoid the closely related lambs, only one individual from the generation was selected. The crossbred lambs (MG×STH) and the STH sheep with a similar birth age (within 3 days, female) were random selected for the present study. In order to identify the molecular markers related to production performance and meat quality for further MAS of sheep, we need selected the individuals that have approximately phenotypes. We have revised it with blue color in the manuscript. Thank you for give us this important suggestion.